# No Access, No Safety: Free Lunch Adversarial Attacks on Black-box NLP Models

## Abstract

Textual adversarial attacks confuse Natural Language Processing (NLP) models, such as Large Language Models (LLMs), by finely modifying the text, resulting in incorrect decisions. Although existing adversarial attacks are effective, they typically rely on knowing the victim model, using extensive queries, or grasping training data, which limits their real-world applications. In situations where there is neither knowledge of nor access to the victim model, we introduce the **Free Lunch Adversarial Attack (FLA)**, demonstrating that attackers can successfully execute attacks armed only with victim texts. To prevent access to the victim model, we create a shadow dataset with publicly available pre-trained models and clustering methods as a foundation for developing substitute models. To address the low attack success rate (ASR) due to insufficient information feedback, we propose the hierarchical substitution model design, generating substitute models that approximate the victim's decision boundaries to enhance ASR. Concurrently, we use diverse adversarial example generation, employing various attack methods to reduce the frequency of model training, balancing effectiveness with efficiency. Experiments with the Emotion and SST5 datasets show that the FLA outperforms existing state-of-the-art methods (+10% ASR) while lowering the attack cost to zero. More importantly, we discover that FLA poses a significant threat to LLMs such as Qwen2 and the GPT family, and achieves the highest ASR of 45.99% even without access to the API, confirming that advanced NLP models still face serious security risks.

## 1 Introduction

Text classification is a task in natural language processing that organizes and categorizes information by automatically assigning text content to predefined categories or labels (Fields et al., 2024).Neural networks have greatly improved the performance of text classification, and these text classification models have become increasingly important, especially after the popularity of large models (LLMs) (Sun et al., 2023). However, textual adversarial attacks trigger significant differences in model outputs by subtly modifying the text, severely affecting the security and stability of the system (Wang et al., 2023; Han et al., 2024). These attacks not only reduce model accuracy, but can also lead to financial losses and legal risks (Fursov et al., 2021).

Existing textual adversarial attacks can be categorized into white-box and black-box attacks (Baniecki & Biecek, 2024). White-box attackers have full access to all model information, including architecture and parameters (Lin et al., 2021). Therefore, the attacker can obtain the gradient information to modify the input text to attack the model. A more practical scenario involves a black-box attack, where the attacker lacks access to the internal workings of the model and can instead perform the attack by either leveraging the training data or observing the model's outputs. The first approach, known as a transfer-based attack, involves using the available training data to train a substitute model and exploiting the transferability of adversarial examples to attack the victim model (Kwon & Lee, 2022). The second approach, referred to as a query-based attack, depends on obtaining the model's output probability scores to optimize the adversarial input in a gradient-free manner. (Hu et al., 2024; Liu et al., 2024; 2023).

Existing NLP models are usually deployed on Web pages through APIs (González-Mora et al., 2023), and attackers cannot directly access the internal structure and parameter information of the models,

making precise and efficient white-box attacks infeasible (Inkawhich et al., 2020; Wu et al., 2021; Yuan et al., 2022; Naseer et al., 2021; Salman et al., 2020; Deng et al., 2021). In addition, for security reasons, such systems generally do not disclose detailed information about training data and outputs (Li et al., 2020; Sun et al., 2022; Fang et al., 2022; Jang et al., 2022; Xie et al., 2019), and probability scores (Liu et al., 2024), to the user, and even prevent malicious cost by limiting the access frequency, thus further increasing the difficulty of black-box attacks (Hu et al., 2024). To address this realistic challenge, we propose a novel attack hypothesis, *i.e.*, can adversarial attacks still be effectively realized in the acknowledge of only the victim text?

We propose a real-world scenario-oriented attack method called the **Free Lunch Attack** (FLA). The attack does not require knowledge of the structure of the victim model, training data, or even access to the victim model, and can be implemented relying only on the victim text. To achieve *zero access to the victim model*, we utilize publicly available pre-trained models to transform the victim text into vector representations and generate pseudo-labels through clustering to construct shadow datasets to train substitute models. However, due to the incomplete accuracy of the labeling of the shadow dataset, the substitute model obtained from the training is not sufficient to generate effective adversarial examples. To enhance the *effectiveness of the attack*, we focus on balancing attack success rate (ASR) and operational efficiency. The Hierarchical Substitution Model Design incrementally aligns substitute models with the victim's decision boundaries, enhancing ASR by refining the match over multiple iterations. Additionally, our Diverse Adversarial Example Generation strategy utilizes multiple attack methods to generate adversarial examples, minimizing the need for frequent retraining of models. This optimized process ensures a more efficient use of resources while maintaining high attack success.

In the text classification task, FLA is able to easily achieve an attack success rate (ASR) of more than 40% at zero query cost, significantly undermining the predictive accuracy of the model. More importantly, FLA not only helps existing attack methods to improve the ASR by 34.99% on average without additional queries, but also poses a far-reaching threat to mainstream Large Language Models (LLMs), such as Qwen2, and the ChatGPT family, which can achieve up to 45.99% ASR. In addition, we deeply explore the attack efficacy of FLA with very few victimized samples or severe inconsistency with the training data distribution, and show that even under these extreme conditions, FLA maintains an ASR more than 27.35%. This demonstrates the highly destructive and indiscriminate nature of the attack against LLMs in real scenarios. Our main **contributions** can be summarized as follows:

- We propose a novel attack scenario for text classification tasks, revealing the possibility that an attacker can exploit potential vulnerabilities to carry out an attack without accessing the model and without knowledge of its structure and training data.

- We design the Free Lunch Attack (FLA) framework, which successfully achieves an efficient attack on the model in a completely black-box environment by constructing a shadow training set and introducing gradient diversity.

- Experiments show that even the current state-of-the-art LLMs, such as Qwen2, and the ChatGPT family, are not able to effectively defend against such attacks, highlighting the profound threat that FLA poses to model security.

## 2 RELATED WORK

**Text White-Box Attack:** In a text white-box attack, attackers exploit complete knowledge of the victim model's internal structure and parameters to optimize input text, generating adversarial examples that mislead the model. For example, FD (Papernot et al., 2016b) generates adversarial examples by perturbing words that have a significant impact on the model's gradient. Similarly, Hotflip (Ebrahimi et al., 2018) iteratively replaces individual words based on their importance score, calculated through gradient computation. Furthermore, TextBugger (Ren et al., 2019) perturbs both characters and words using a greedy algorithm to maximize the disruption of the gradient. However, the reliance of white-box attacks on full access to the model limits their applicability in real-world scenarios, where such information is typically unavailable.

**Text Black-Box Attack:** In text black-box attack, attackers generate adversarial examples without access to the model's internal architecture or parameters, relying solely on input-output observations. Existing black-box text adversarial attack methods can be categorized into query-based attack and

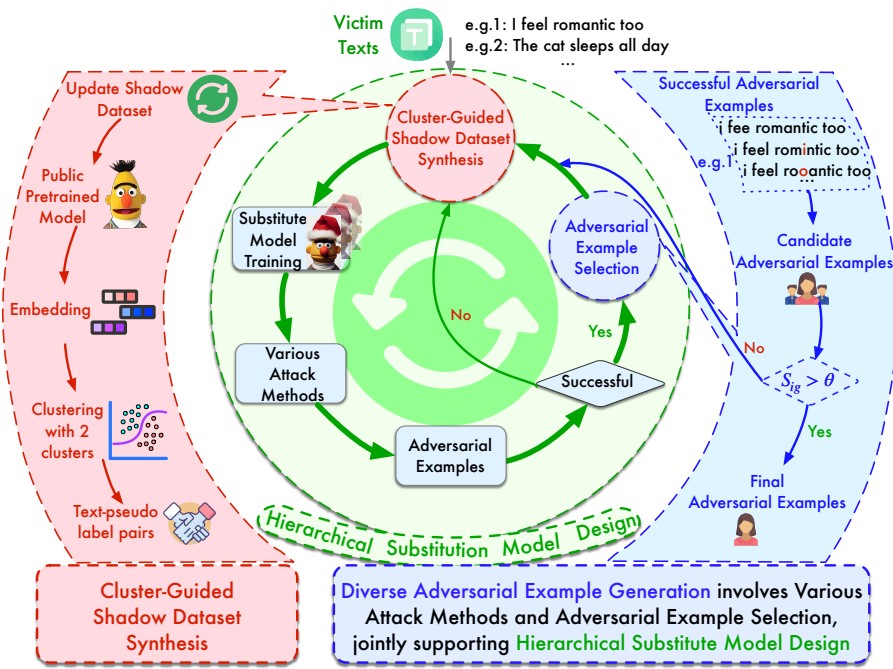

Figure 1: **The Overview of FLA**. FLA utilizes a cluster-guided shadow dataset synthesis to create text-cluster label pairs for substitute model training. It then applies various attack methods to generate adversarial examples. If all adversarial examples fail to attack the substitute model, a new substitute model is trained to produce new adversarial examples. For successful attacks, if the maximum attack score $S_{ig}$ exceeds the threshold $\theta$, the highest-scoring example is chosen as the final adversarial example; otherwise, a new substitute model is required to generate new adversarial examples.

transfer-based attacks (Waghela et al., 2024; Han et al., 2024; Zhu et al., 2024; Kang et al., 2024). In a query-based attack, the attacker can determine the importance of words based on the probability scores of the model's output labels and target the important words sequentially until the victim model generates a different label. SememePSO (Zang et al., 2020) uses a metaheuristic approach to optimize the search space for generating adversarial examples. BAE (Garg & Ramakrishnan, 2020) replaces words based on the BERT model and prompt learning. Leap (Xiao et al., 2023) combines Levy flight initialization with adaptive particle swarm optimization and speeds up convergence via greedy mutation. HQA (Liu et al., 2024) iteratively substitutes words to minimize perturbation by selecting optimal synonyms. In transfer-based attack, attackers generate adversarial examples on the substitute model and uses them to attack the victim model by exploiting similarities between the models. For example, CT-GAT (Lv et al., 2023) develops a sequence-to-sequence generative model by leveraging adversarial data from various tasks, learning general adversarial features to produce adversarial examples across different tasks. However, real-world systems typically hide detailed information about probability scores, substitute model and limit access frequency to prevent malicious exploitation, marking new challenges to black-box attacks.

In summary, the need for internal model information makes text white-box attacks challenging to implement in real-world scenarios, while restrictions on probability scores, substitute models, and query limits may hinder the effectiveness of black-box attacks. In contrast, FLA is well-suited for more practical attack environments, as it can generate adversarial examples with high success rates even when only the victim texts or victim texts' topic is available. .

## 3 PRELIMINARY

❶ **Victim Model**: The **victim model** refers to the target model that the attacker intends to attack, denoted as $f_v$. ❷ **Victim Text (Test Set)**: The victim text (test set) refers to the text that the attacker intends to modify. This is the text that the attacker is inevitably able to access. ❸ **Substitute Model**:

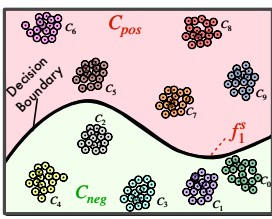 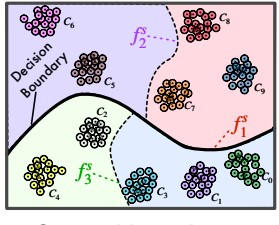 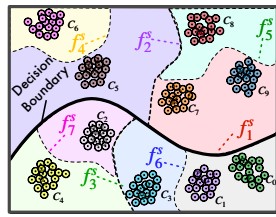

| **First Iteration** | **Second Iteration** | **Third Iteration** |

Figure 2: **The Overview of Hierarchical Substitution Model Design.** For the first iteration of FLA, all victim texts are employed to train the substitute model $f_1^s$. For the second iteration, the victim texts with the positive cluster label $C_{\text{pos}}$ are employed to train the substitute model $f_2^s$, the victim texts with the negtive cluster label $C_{\text{neg}}$ are employed to train the substitute model $f_3^s$. Following this process, we can train the hierarchical substitution models.

The substitute model approximates the decision boundary of the victim model and is used to generate adversarial samples, denoted as $f_s$ and $f_s(x) \approx f_v(x)$. ❹ **Shadow Dataset**: The shadow dataset refers to data that are accessible to the attacker. In victim text-only scenarios, the shadow dataset typically consists of unlabeled victim texts, such as those found in the test set. However, the shadow dataset does not necessarily need to be the test set; it only needs to share similar attributes. For example, in the 5-class sentiment analysis dataset SST5, the shadow dataset could be the unlabeled SST5 test set or an online sentiment dataset, like the binary Tomatoes dataset. As a result, the labels and data distribution in the shadow dataset may differ from those in the training dataset. ❺ **Adversary's Goals** Consider a text classification model $f_v$ that predicts labels $\hat{y} = [C_1, C_2, \ldots, C_m]$, where $m$ represents the total number of labels. At the same time, attackers only have access to victim texts, denoted $\{x_1, x_2, \ldots, x_n\}$, where $n$ corresponds to the number of victim texts. Our goal is to generate adversarial examples $\{\tilde{x}_1^*, \tilde{x}_2^*, \ldots, \tilde{x}_n^*\}$ just based on victim texts $\{x_1, x_2, \ldots, x_n\}$ and to successfully fool the victim model $f_v$, which aims at $f_v(x_i) \neq f_v(\tilde{x}_i^*)$.

## 4 FREE LUNCH ADVERSARIAL ATTACK

To carry out the NLP model attack without access and understanding, we propose the Free Lunch Attack (FLA), which creates a shadow dataset by generating cluster labels using K-means (Lloyd, 1982) clustering to achieve zero-access to the victim model. Next, we use the text-cluster label pairs to train the substitute model. Various attack methods are applied to generate adversarial examples. If all adversarial examples fail against the substitute model, a new substitute model is trained using the hierarchical substitution model design process. For the successful adversarial examples, we compute their attack scores. If the maximum attack score $S_{ig}$ exceeds the threshold $\theta$, the highest-scoring example is selected as the final adversarial example. Otherwise, a new substitute model, trained using the hierarchical substitution model design, is required to generate additional adversarial examples.

### 4.1 CLUSTER-GUIDED SHADOW DATASET SYNTHESIS

Training a substitute model without access to the original training data poses a significant challenge in adversarial settings. This section proposes a method to synthesize a shadow training dataset by assigning pseudo-labels to victim texts.

We find that the victim data and training data exhibit similar distribution characteristics, a common supposition in machine learning suggesting that training and test sets follow the same distribution Zhu et al. (2019). Building on this, we hypothesize that texts eliciting strong responses from the model are likely akin to the training data, offering a viable proxy for estimating the training dataset.

To utilize victim texts effectively, we apply a clustering method to assign pseudo-labels to these unlabeled texts, thus creating a shadow dataset. As the Figure 1 shows, each victim text $x_i$ is embedded using a pre-trained model $f_{\text{pre}}$, denoted as $\boldsymbol{E}(x_i) = f_{\text{pre}}(x_i)$. All victim text embeddings $\boldsymbol{E} = [\boldsymbol{E}(x_1), \boldsymbol{E}(x_2), \ldots, \boldsymbol{E}(x_n)]$ are then clustered into distinct groups, and cluster labels $y_i^{\text{clu}}$ are assigned as pseudo-labels for each text. The resulting shadow dataset $\boldsymbol{D}$ is defined as:

$$\boldsymbol{D} = \{(x_1, y_1^{\text{clu}}), (x_2, y_2^{\text{clu}}), \ldots, (x_n, y_n^{\text{clu}})\} \tag{1}$$

This dataset effectively approximates the distribution of the original training data, facilitating the training of the substitute model. Despite the discrepancies between victim data-pseudo label pairs and the original data-label pairs, which can hinder perfect alignment of the substitute model with the victim model's decision boundary, our aim is not to mirror the victim model but to develop a substitute model that can distinguish between classes clearly. The substitute model, trained on the shadow dataset $\boldsymbol{D}$, while not an exact replica of the victim model, is adequate for conducting effective adversarial attacks.

## 4.2 HIERARCHICAL SUBSTITUTION MODEL DESIGN

The substitute model trained in Section 4.1 can clearly distinguish between classes, but it is not similar to the victim model, which compromises the effectiveness of the transfer attack. To address this, we propose a hierarchical training approach that aims to align substitute models more closely with the victim model's adversarial decision boundary.

Multiple substitute models are used to incrementally generate adversarial decision boundaries that progressively converge towards the victim model's boundary, enhancing the likelihood of a successful attack (Papernot et al., 2016a). This progressive alignment enhances the probability of a successful attack. Theorem 1 formalizes this intuition.

**Theorem 1.** *As the number of substitute models $m$ increases, the probability of successfully attacking the victim model, denoted by $p_m^{suc}$, also increases, meaning:*

$$\text{for } m > n, \quad p_m^{suc} > p_n^{suc}. \tag{2}$$

*Proof.* Please see Appendix E for the proof.

The process starts by training a substitute model, $f_1^s$, using a shadow dataset $\boldsymbol{D}$. Then, the shadow dataset is updated to train a new substitute model. As the Figure 2 shows, the whole process is as follows: ❶ For each victim text $x_i$, we check its cluster label $y_i^{\text{clu}}$. If this label corresponds to the positive cluster $C_{\text{pos}}$, we include $x_i$ in a new set of victim texts. ❷ After this, we obtain a new set of victim texts $\{x_1^{\text{new}}, x_2^{\text{new}}, \ldots, x_{n_1}^{\text{new}}\}$, where $n_1 < n$, and each text has the positive cluster label $C_{\text{pos}}$. ❸ Next, we perform a cluster-guided shadow dataset synthesis on this new set, assigning new cluster labels $y_i^{\text{clu,new}}$ to each victim text $x_i^{\text{new}}$. ❹ We update the shadow dataset $\boldsymbol{D}_{\text{new}}$, which contains these newly labeled texts, and then use $\boldsymbol{D}_{\text{new}}$ to train a new substitute model, $f_1^s$. Similarly, for texts with negative cluster labels, we repeat this process to train another substitute model, $f_2^s$.

By repeating this process $u$ times, we generate $U$ substitute models, where the total number of models is as follows:

$$U = 2^0 + 2^1 + 2^2 + \cdots + 2^u. \tag{3}$$

Now, for each victim text $x_i$, we apply the diverse adversarial example generation (Section 4.3) to create an adversarial example $\tilde{x}_i^1$. If this example fails to fool the current substitute model $f_1^s$ (*i.e.*, $f_1^s(x_i) = f_1^s(\tilde{x}_i^1)$), we move on to the next substitute model, $f_2^s$, and attempt to create a new adversarial example. This process continues until the adversarial example successfully attacks a substitute model, meaning $f_j^s(x_i) \neq f_j^s(\tilde{x}_i^1)$. The goal is to keep trying different substitute models until the attack is successful.

## 4.3 DIVERSE ADVERSARIAL EXAMPLE GENERATION

Training substitute models can be resource-intensive, especially when repeated iterations are required to generate effective adversarial examples. In each iteration, a new substitute model is trained on a subset of the shadow dataset, which is progressively reduced, limiting the available data for further training. Consequently, the continuous training of new substitute models becomes less feasible and increasingly costly. To address this, it is essential to evaluate the necessity of training additional substitute models and explore methods that minimize the associated training costs.

This section proposes a strategy to reduce training expenses by utilizing diverse adversarial examples generated from multiple attack methods. Using multiple attack methods improves the chance of

successfully attacking substitute models compared to relying on a single method. This reduces the costs of repeatedly training substitute models.

Consider $w$ attack methods $\{M_1, M_2, \ldots, M_w\}$, each with success probabilities $\{p_1, p_2, \ldots, p_w\}$. The overall success probability is $\max\{p_1, p_2, \ldots, p_w\}$, which increases as more attack methods are employed. Theorem 2 formalizes this intuition.

**Theorem 2.** *Suppose that $\{p_1, p_2, \ldots, p_w\}$ be independent and identically distributed (i.i.d.) success probabilities with cumulative distribution function (CDF) denoted by $F(p)$. Then as $w \to \infty$, the probability that $\max\{p_1, p_2, \ldots, p_w\} > p_i$ approaches 1 for any $p_i < 1$. Formally,*

$$\lim_{w \to \infty} \Pr\left(\max\{p_1, p_2, \ldots, p_w\} > p_i\right) = 1,$$

$$and \quad \lim_{w \to \infty} \Pr\left(\max\{p_1, p_2, \ldots, p_w\} > \max\{p_1, p_2, \ldots, p_m\}\right) = 1, \quad for\ w > m. \tag{4}$$

*Proof.* Please see Appendix F for the proof.

Theorem 2 shows that as $w \to \infty$, the probability of success approaches 1, indicating the benefit of using multiple attack methods to reduce training costs. Take the substitute model $f_1^s$ as an example, if an adversarial example $\tilde{x}_i^1 = M_1(x_i, f_j^s)$ fails to fool the model (i.e., $f_1^s(x_i) = f_1^s(\tilde{x}_i^1)$), this suggests a potential issue with either the one attack method $M_1$ or the substitute model. Therefore, before deciding to train a new substitute model, we eliminate the influence of the failed attack method by employing multiple attack methods.

Specifically, we use $w$ attack methods $\{M_1, M_2, \ldots, M_w\}$ to generate $w$ adversarial examples:

$$\{\tilde{x}_i^1, \tilde{x}_i^2, \ldots, \tilde{x}_i^w\}, \quad \text{where} \quad \tilde{x}_i^j = M_j(x_i, f_j^s). \tag{5}$$

If none of these adversarial examples successfully fool $f_1^s$, i.e., $f_1^s(x_i) = f_1^s(\tilde{x}_i^j)$ for all $j$, a new substitute model must be trained.

Among the successful adversarial examples, we select the one with the best attack performance, measured by the logit change and the similarity between $x_i$ and the adversarial example $\tilde{x}_i^j$. The attack performance score is:

$$s_{ij} = \alpha \left(o_{\hat{y}_i}(x_i) - o_{\hat{y}_i}(\tilde{x}_i^j)\right) + \beta \left(\frac{\boldsymbol{E}(x_i) \cdot \boldsymbol{E}(\tilde{x}_i^j)}{\|\boldsymbol{E}(x_i)\| \cdot \|\boldsymbol{E}(\tilde{x}_i^j)\|}\right), \tag{6}$$

where $\alpha$ and $\beta$ are scaling factors, and $\boldsymbol{E}(x_i)$ and $\boldsymbol{E}(\tilde{x}_i^j)$ are the embeddings. The adversarial example with the highest score $s_{ij}$ is selected as $\tilde{x}_i^g$. If this score exceeds a threshold $\theta$, $\tilde{x}_i^g$ is accepted as the final adversarial example. Otherwise, a new substitute model is trained to generate a new adversarial example.

## 5 EXPERIMENT

### 5.1 EXPERIMENTAL SETUP

❶ **Datasets:** We perform primary experiments on the SST5 and Emotion datasets. The details of the datasets are presented in Appendix B. ❷ **Metrics:** Several metrics are used to evaluate the effectiveness of FLA, including Attack Success Rate (ASR), Cost, and Semantic Similarity (Sim) between the original texts and adversarial examples. We utilize the metric "Cost" as it more accurately reflects real-world application scenarios. In this paper, "Cost" is defined as the number of queries made by the attack method to the model, measured in units of "tokens." Detailed descriptions of these evaluation metrics are provided in Appendix C. ❸ **Substitute model:** The details of the substitute model are presented in Appendix D. ❹ **Baselines:** In the victim texts-only scenario, no prior work on attacks has been carried out. The stringent constraints of this scenario render all existing text attack methods ineffective. We have broadened the application conditions of other methods, enabling other text-attack algorithms to access any required information. Sevel text attack methods are selected, including Bae (Garg & Ramakrishnan, 2020), FD (Papernot et al., 2016b), Hotflip (Ebrahimi et al., 2018), PSO (Zang et al., 2020), TextBug (Ren et al., 2019), Leap (Xiao et al., 2023), CT-GAT (Lv et al., 2023), and HQA (Liu et al., 2024). The detail of these methods are presented in Appendix

Table 1: The attack performance of FLA and adversarial attacks on Emotion and SST5 datasets. For each metric, the best method is highlighted in **bold** and the runner-up is underlined.

| Method | SST5 | | | | | | Emotion | | | | | |
|---|---|---|---|---|---|---|---|---|---|---|---|---|
| | DistilBERT | | | RoBERTa | | | DistilBERT | | | RoBERTa | | |
| | ASR(%) ↑ | Sim ↑ | Cost ↓ | ASR(%) ↑ | Sim ↑ | Cost ↓ | ASR(%) ↑ | Sim ↑ | Cost ↓ | ASR(%) ↑ | Sim ↑ | Cost ↓ |
| Bae | 42.71 | 0.888 | 47360 | 39.14 | 0.887 | 47466 | 32.25 | 0.926 | 43682 | 32.95 | 0.923 | 43656 |
| FD | 25.20 | 0.939 | 27760 | 22.30 | **0.982** | 21452 | 22.30 | 0.932 | 25612 | 27.50 | **0.982** | 22850 |
| Hotflip | 41.50 | 0.951 | 25455 | 29.00 | 0.951 | 25956 | 29.00 | 0.949 | 28566 | 28.05 | 0.949 | 28800 |
| PSO | 45.14 | 0.954 | 24398 | 41.50 | 0.954 | 27360 | 39.50 | 0.952 | 23660 | 37.65 | 0.951 | 24190 |
| TextBug | 30.36 | **0.978** | 69520 | 20.85 | 0.978 | 67009 | 20.85 | **0.978** | 60642 | 21.45 | 0.978 | 60662 |
| Leap | 32.55 | 0.953 | 21548 | 30.07 | 0.944 | 21083 | 40.58 | 0.926 | 19460 | 37.63 | 0.911 | 19560 |
| CT-GAT | 29.37 | 0.939 | 46238 | 24.80 | 0.926 | 82957 | 28.10 | 0.904 | 52114 | 30.85 | 0.906 | 50686 |
| HQA | 46.11 | 0.936 | 64855 | 39.64 | 0.929 | 64256 | 37.40 | 0.912 | 44876 | 36.40 | 0.911 | 46326 |
| **FLA** | **52.08** | 0.950 | **0** | **45.03** | 0.950 | **0** | **43.15** | 0.949 | **0** | **42.05** | 0.949 | **0** |

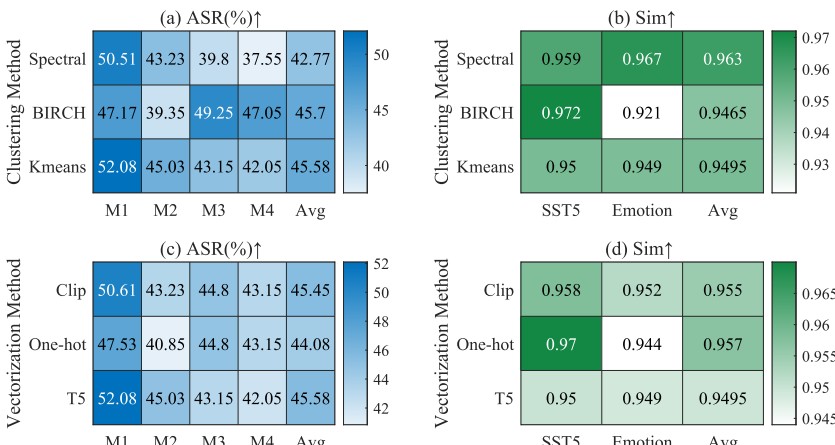

Figure 3: Experiment results under different clustering vectorization method. M1 and M2 represent DistilBERT and RoBERTa models for SST5, while M3 and M4 represent DistilBERT and RoBERTa models for Emotion. "Avg" denotes the mean score of ASR and similarity.

A. To ensure a fairer comparison, we restrict other attack methods to a maximum of 35 accesses per victim text, as FLA does not access the victim model at all. ❺ **Other Setup:** The pre-trained embedding model "T5" (Raffel et al., 2020) is effectively used to embed the victim texts and calculate their similarity. Furthermore, we apply the K–means clustering method to categorize the embedded texts. The scaling factors $\alpha$ and $\beta$ in equation 6 are set to 9 and 1, respectively. And the threshold $\theta$ is set to 4

## 5.2 COMPARISON WITH OTHER ADVERSARIAL ATTACKS

Table 1 presents the experimental results for both the SST5 and Emotion datasets. Our FLA method achieves state-of-the-art (SOTA) performance in both ASR and Cost metrics, demonstrating superior attack success rates with fewer queries. Notably, FLA delivers these results without querying the victim model or relying on any information regarding the model, ground truth, or training data. Regarding the similarity metric, while semantic similarity may be somewhat compromised during the hierarchical substitution model process, our diverse adversarial example generation method allows us to maintain a relatively high level of semantic similarity. Additionally other attack algorithms can improve ASR by increasing query costs, but this improvement comes at the expense of a significant number of queries, making it impractical for real-world applications.

## 5.3 ABLATION STUDY

**Different Clustering Method:** In this paper, the FLA method utilizes K-means as its clustering method. To mitigate the influence of different clustering methods, we introduce additional clustering methods, including Spectral clustering  (Von Luxburg, 2007)and BIRCH (Zhang et al., 1996). The

Table 2: Few-shot attack performance of FLA on Emotion and SST5 datasets.

| Shot Size | Emotion | | | SST5 | | |
|---|---|---|---|---|---|---|
| | Roberta ASR(%) ↑ | Distilbert ASR(%) ↑ | Sim↓ | Roberta ASR(%) ↑ | Distilbert ASR(%) ↑ | Sim↓ |
| 1-shot | 27.55 | 27.35 | 0.929 | 39.28 | 48.01 | 0.901 |
| 2-shot | 29.75 | 30.00 | 0.923 | 41.27 | 50.14 | 0.897 |
| 4-shot | 30.15 | 29.30 | 0.923 | 37.51 | 46.06 | 0.902 |
| 8-shot | 33.25 | 33.30 | 0.916 | 38.51 | 46.61 | 0.905 |
| Full-shot | 42.05 | 43.15 | 0.949 | 45.03 | 52.08 | 0.950 |

results are presented in Figure3. Subplots (a) and (b) of Figure3 display the ASR and similarity metrics for various clustering methods. While the methods exhibit different levels of ASR and similarity between victim texts and models, the differences are not significant. Furthermore, no single method consistently achieves SOTA performance across all scenarios. These findings suggest that, although clustering methods can influence the effectiveness of the attack, the overall impact remains limited. **Different Vectorization Methods:** The "T5" pre-trained model is the vectorization method in FLA. To mitigate the influence of different vectorization methods, we introduce additional vectorization methods, including CLIP (Radford et al., 2021) and one-hot encoding (Rodríguez et al., 2018). CLIP is a pre-trained model trained on an image-text pair dataset. In contrast, a one-hot encoding represents categorical data using a binary vector where an element is set to 1 and all others are set to 0. Unlike CLIP, one-hot encoding is not pre-trained and does not rely on any training data. The results are presented in Figure3. The subplots (c) and (d) of Figure3 present the ASR and similarity of various vectorization methods. Although the methods show different levels of ASR and similarity between victim texts and models, the differences are not substantial. However, no single method consistently achieves SOTA performance in all scenarios.

These results indicate that *while different vectorization methods influence the effectiveness of the attack, the overall impact is not substantial.* **Different Cluster Number:** Based on the analysis in Section 4.1, using two clusters yields the optimal attack performance. Therefore, FLA implements two clusters in its approach. To evaluate the impact of different cluster numbers on attack performance, we conduct experiments with two, three, and four clusters. The primary evaluation metric is the Attack Success Rate (ASR). As shown in Figure 4, the success rate decreases as the number of clusters increases. *The highest success rate occurs with two clusters, validating our choice for FLA.*

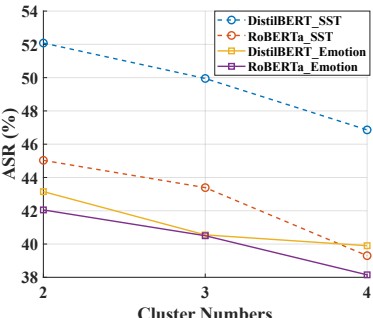

Figure 4: The ASR for different cluster numbers. Fewer clusters imply better ASR.

## 5.4 REAL-WORLD ATTACK

### 5.4.1 FEW-SHOT VICTIM TEXTS

In few-shot victim texts scenario, attackers only acquire a minimal portion of the victim texts. Specifically, for each label in the SST5 and Emotion datasets, we select 1, 2, 4, and 8 victim texts respectively. The ASR results for the few-shot data are presented in Table 2. Even when each label is associated with only one text, the FLA algorithm still achieves ASR values of 27.55%, 27.35%, 39.28%, and 48.01%, respectively. Notably, in the SST5 dataset, a decrease in the number of victim texts does not significantly reduce the ASR. Notably, with very few samples, fewer samples can sometimes yield better attack results. This indicates that when the sample size is extremely small, the ASR fluctuates significantly. Meanwhile, a decrease in the number of victim texts leads to a reduction in similarity; however, this decline progresses at a relatively slow rate. Our results indicate that even with a very limited amount of victim text, FLA can successfully attack the victim model.

### 5.4.2 ZERO-SHOT VICTIM TEXTS

In previous experiments, we assume that the attacker can access to the victim texts. However, we now investigate a scenario known as the **zero-shot data** scenario, in which the attacker does not

Table 3: Zero-shot attack performance of FLA. The attackers can access the Go-emotion or Rotten Tomatoes datasets and apply the FLA method to craft adversarial examples targeting the SST5 and Emotion datasets.

| Victim Dataset | Access Dataset | Roberta ASR(%) ↑ | Distilbert ASR(%) ↑ | Sim↓ |
|---|---|---|---|---|
| SST5 | Go-emotion | 39.46 | 50.36 | 0.911 |
| | Tomatoes | 40.00 | 49.82 | 0.909 |
| | SST5 | 45.03 | 52.08 | 0.950 |
| Emotion | Go-emotion | 30.15 | 33.60 | 0.932 |
| | Tomatoes | 31.05 | 32.45 | 0.927 |
| | Emotion | 42.05 | 43.15 | 0.949 |

have access to the victim texts. In this case, although the attacker lacks direct access to the victim texts, they are aware of key attributes. Specifically, the attacker knows that the victim texts are related to an emotion classification task. Leveraging this information, the attacker gathers similar sentiment analysis datasets from the Internet, such as Go-Emotions (Demszky, 2020) and Rotten Tomatoes (Pang & Lee, 2005). We then apply FLA to these datasets to train the substitute model. The results, presented in Table 3, show that without access to the victim texts or model, FLA still achieves an ASR of over 30%. For example, on the SST5 dataset, when DistilBERT is used as the victim model and Go-emotion data is applied, the ASR even reaches 50.36%, almost identical to the 50.36% achieved using SST5 victim texts directly. In the zero-shot data scenario, although the ASR decreases, it remains significantly high. However, compared to the ASR, the zero-shot data scenario exerts a more pronounced negative effect on similarity. In summary, our findings suggest that *an attack remains viable even without direct access to the victim texts, as long as the datasets used share similar attributes.*

### 5.4.3 LLMs Attack

We also discuss the effectiveness of FLA in LLMs, including Qwen2 (Yang et al., 2024), ChatGPT4o and ChatGPT4omini (OpenAI, 2023), by comparing the effectiveness of FLA and transfer attacks. We construct agents using LLMs through prompt learning. Below is our prompt template, using the Emotion dataset as an example: prompt "The objective is to predict the label of the provided text. It is sufficient to supply the label alone. The labels encompass 'Anger', 'Fear', 'Joy', 'Love', 'Sadness', and 'Surprise', excluding any other classifications." In closed-source LLMs and prompt-based learning scenarios, where gradients and label probabilities are unavailable, our baseline shows that CT-GAT is capable of launching an attack. Consequently, we compare the attack performance of CT-GAT with that of FLA. Table 5 displays the experimental results for LLMs. Surprisingly, FLA achieves an attack success rate of over 27% across all datasets and state-of-the-art LLMs, even with only limited access to the victim texts.These results indicate that an attacker can successfully target closed-source large language models with access to only the victim texts.

### 5.5 Other experiments

**The Impact of Substitute Model Quantity on Attack Effectiveness**: To examine the effect of the number ofsubstitute models on attack performance, we created scenarios using 1, 3, and 7 substitute models.Furthermore, to eliminate the impact of varying attack algorithms, we applied only a single attack algorithm in each scenario. The results are presented in Table 4, employing more substitute models leads to a gradual increase in ASR, at times even surpassing the performance of directly applying the attack method to the victim model. For example, when the victim texts are SST5 and the victim model is DistilBERT, the TextBug method achieves a 30.36% ASR on the victim model while using TextBug with 7 substitute models

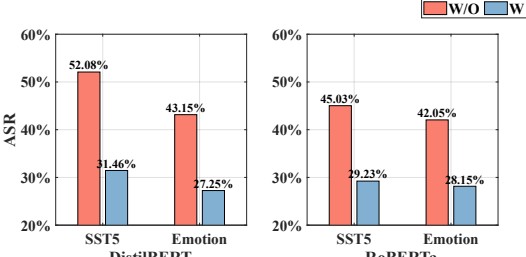

Figure 5: The ASR of FLA with and without the defense method. "W" indicates the FLA with the defense method. And "W/O" indicates the FLA without the defense method.

Table 4: The ASR of FLA's variants with different substitute model quantity . "Bae-FLA", "FD-FLA", "Hotflip-FLA", "PSO-FLA", and "TextBugger-FLA" indicate that transfer attack based on the substitute model generated by FLA method. These attacks are implemented by Bae, FD, Hotflip, PSO, and TextBugger methods.

| Dataset | FLA's Variants | DistilBERT | | | RoBERTa | | |
|---|---|---|---|---|---|---|---|
| | | 1 Model Used | 2 Models Used | 7 Models Used | 1 Model Used | 2 Models Used | 7 Models Used |
| SST5 | Bae-FLA | **35.20** | **40.72** | **46.38** | **36.97** | **41.76** | **49.41** |
| | FD-FLA | 20.54 | 29.91 | 31.27 | 20.50 | 24.21 | 23.71 |
| | Hotflip-FLA | 24.71 | 25.88 | 33.89 | 23.53 | 24.66 | 32.99 |
| | PSO-FLA | 26.06 | 34.30 | 34.12 | 24.16 | 28.28 | 29.23 |
| | TextBug-FLA | 23.03 | 25.43 | 42.22 | 19.41 | 21.09 | 33.54 |
| Emotion | Bae-FLA | **25.55** | **30.80** | **42.50** | **26.60** | **31.40** | **43.25** |
| | FD-FLA | 20.05 | 27.15 | 31.65 | 20.90 | 26.75 | 32.30 |
| | Hotflip-FLA | 19.40 | 19.95 | 28.05 | 17.85 | 18.60 | 26.85 |
| | PSO-FLA | 20.85 | 20.55 | 29.00 | 21.00 | 20.15 | 27.75 |
| | TextBug-FLA | 13.95 | 13.60 | 42.40 | 14.05 | 13.50 | 39.25 |

Table 5: The ASR(%)↑ of LLMs.

| Dataset | SST5 | | | Emotion | | |
|---|---|---|---|---|---|---|
| Victim Model | GPT4o | GPT4omini | Qwen2 | GPT4o | GPT4omini | Qwen2 |
| CT-GAT | 20.23 | 19.85 | 27.36 | 16.15 | 17.85 | 24.20 |
| FLA | 35.37 | 36.91 | 45.99 | 27.65 | 29.30 | 36.35 |

increases the ASR to 42.22%. *The results suggest that increasing the number of substitute models can enhance attack performance.* **Attack Results under Defense Method**: Adversarial training is a widely used defensemechanism (Qiu et al., 2019); however, defensemechanism (Qiu et al., 2019); however, retrainingmodels using this approach is resource-intensive. Therefore, training-free defense methods are more appropriate for FLA. In this work, we adopt the defense method proposed by (Wang et al., 2023) and apply prompt learning on large language models (LLMs) to mitigate adversarial text inputs. As shown in Figure 5, the attack success rate decreases significantly after applying this defense, though the attack remains partially effective. **Further Experimental Results on Additional Datasets**: To further explore the attack effectiveness of FLA, we conducted experiments on two additional datasets: Agnews (Zhang et al., 2015) and TREC6 (Voorhees & Harman, 2000). FLA demonstrated strong attack performance across both datasets. Detailed results are presented in Table 7 in appendix G, FLA achieves zero queries, the best results among all approaches. In DistilBert victim model, FLA's ASR outperformed the second-best method by 4% on Agnews, and 8.08% on TREC6. When RoBERTa is used as the victim model, FLA also achieves SOTA attack results. These results demonstrate that FLA effectively attacks a diverse range of datasets.

## 6 CONCLUSION

In this paper, we present a novel scenario called the victim texts-only scenario and the corresponding approach, referred to as FLA. Our study is the first to demonstrate that text adversarial attacks do not require information about the victim model, labeling, or auxiliary data. Attackers can generate adversarial examples using only the victim texts. Additionally, both few-shot and zero-shot scenarios show that adversarial examples can still be generated, even with limited victim texts or minimal knowledge of their attributes. To the best of our knowledge, the victim texts-only scenario represents the most stringent attack scenario for adversarial examples. If an attack method is successful in this scenario, it is likely to succeed in other attack scenarios as well. This is because the victim-only text scenario requires the least amount of information, making it the most challenging. In contrast, other scenarios provide more data, potentially making them easier to exploit. We are also very excited about the future applications of FLA in other modalities in future work.

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

This appendix includes our supplementary materials as follows:

- More details of baselines in Section A

- More detail of dataset in Section B

- More detail of evaluate metrics in Section C

- More detail of substitute model in Section D

- More mathematical proofs of Theorem 1 in Section E

- More mathematical proofs of Theorem 2 in Section F

- More results on additional datasets in Section G

Table 6: The statistics of datasets.

| Dataset | Train | Test | Type | Number of labels | Labels name |
|---------|-------|------|------|------------------|-------------|
| Tomatoes | 8530 | 1066 | Sentiment | 2 | Positive, Negtive |
| SST5 | 8544 | 2210 | Sentiment | 5 | Very positive, Positive, Neutral, Negative, Very negative |
| Go-emotions | 43410 | 5427 | Sentiment | 28 | Admiration, Amusement, Anger, ... , Surprise, Neutral |
| Emotion | 16000 | 2000 | Sentiment | 6 | Sadness, Joy, Love, Anger, Fear, Surprise |

## A  THE DETAILS OF BASELINES METHODS

In this section, we present the details of the attack methods we used, including Bae, FD, Hotflip, PSO and TextBug **Bae**: Bae (BERT-based Adversarial Examples) is an advanced attack method utilizing the BERT pre-trained model and prompt learning. This approach involves systematically replacing words in the input text to create adversarial examples, thereby testing the robustness of natural language processing models.

**FD**: The FD (Frequency Domain) attack method replaces words with synonyms based on the gradient descent optimization of the victim model. This technique aims to subtly alter the input text to generate adversarial examples while maintaining semantic coherence.

**Hotflip (Ebrahimi et al., 2018)**: Hotflip (Ebrahimi et al., 2018) is a text white-box attack method. It iteratively replaces individual words based on their calculated importance. The importance of each word is determined by the magnitude of the gradient in the victim model, allowing for precise identification of the most impactful words to alter.

**PSO**: PSO is a text soft-label black-box attack method. It employs a versatile metaheuristic approach to optimize the search space for generating adversarial examples, thereby enabling the victim model to produce varied outputs. This method leverages the sememe-based representations of words to effectively navigate and perturb the input text.

**TextBug**: TextBug is a versatile attack framework designed to generate adversarial text against real-world applications. It operates in both white-box and black-box settings by perturbing words based on their significance. In the white-box setting, it uses the Jacobian matrix to identify critical words and applies various perturbations. In the black-box setting, it employs a scoring function to determine word importance. TextBug is effective and efficient, preserving the text's original utility while achieving high success rates in misleading state-of-the-art NLP systems.

## B  THE DETAILS OF DATASETS

In this section, we present the details of the datasets we used, including Tomatoes, Emotion, Go-emotion, and SST5 datasets. The resuls is presented in Table 6. The **Emotion** dataset, which encompasses six distinct Emotions, is derived from Twitter messages. **SST5** The SST-5 dataset, a sentiment analysis resource comprising five categories, originates from movie reviews.

## C  THE DETAILS OF EVALUATION METRICS

**Attack Success Rate:** Attack Success Rate is calculated as the ratio of the number of success attack adversarial examples to the total number of all adversarial examples. The higher ASR signifies the better attack method. **Number of Cost:** This metric denotes the quantity of Cost that attackers direct towards the victim model. A reduced query count suggests a more efficient attack method. **Semantic Similarity:** Semantic Similarity is assessed by computing the mean similarity between the perturbed texts and the original texts. An elevated Semantic Similarity implies a more potent attack strategy.

## D  THE DETAILS OF SUBSTITUTE MODEL

**Experiment Setting:** The substitute model is developed utilizing a transformer-based architecture, which serves as the fundamental backbone of this model. This architecture incorporates 12 hidden layers, each with a size of 768 . The dropout probability is set at 0.1 . The model is trained using the AdamW optimizer. The training process employs a batch size of 32 , a learning rate o0.00005, and is carried out over 2 epochs. Our substitute model consists of 12 transformer blocks, each with 768 hidden units and 12 self-attention heads. Each transformer block contains the following substructures:

- **Self-Attention Layer:** The hidden size of the self-attention layer is 768.
- **Position-wise Feed-Forward Networks:** This network first maps the output of the attention layer to a 3072-dimensional feature space through a fully connected layer, then applies a ReLU activation function for non-linear activation, and finally maps the 3072-dimensional feature space back to a 768-dimensional feature space through a second fully connected layer.
- **Layer Normalization and Residual Connection:**
    - **Layer Normalization:** Applied to the output of each sub-layer to stabilize the training process.
    - **Residual Connection:** Adds the normalized output to the input of the sub-layer.

## E  THE PROOF OF THEOREM 1

*Proof.* Consider $m$ substitute models $S_1, S_2, \ldots, S_m$ used to generate adversarial examples against the victim model $V$. Let the probability of successfully attacking the victim model using adversarial examples from substitute model $S_i$ be denoted by $p(S_i \to V)$. The overall probability of success using $m$ substitute models, denoted as $p_m^{\text{suc}}$, can be expressed as:

$$p_m^{\text{suc}} = 1 - \prod_{i=1}^{m} \left(1 - p(S_i \to V)\right),$$

where $\prod_{i=1}^{m}(1 - p(S_i \to V))$ represents the probability of none of the substitute models succeeding in attacking the victim model. Thus, the complement gives the overall success probability of at least one substitute model generating a successful adversarial attack.

Now, consider the scenario where the number of substitute models is increased from $n$ to $m$, with $m > n$. Let the additional substitute model be denoted as $S_{m+1}$, with its success probability given by $p(S_{m+1} \to V)$. The overall success probability with $m + 1$ substitute models is given by:

$$p_{m+1}^{\text{suc}} = 1 - \prod_{i=1}^{m+1} \left(1 - p(S_i \to V)\right).$$

Expanding this, we obtain:

$$p_{m+1}^{\text{suc}} = 1 - \prod_{i=1}^{m} \left(1 - p(S_i \to V)\right) \cdot \left(1 - p(S_{m+1} \to V)\right).$$

Since $0 < p(S_{m+1} \rightarrow V) < 1$, it follows that $1 - p(S_{m+1} \rightarrow V) < 1$. Therefore, multiplying by $(1 - p(S_{m+1} \rightarrow V))$ reduces the overall probability of failure, which implies that the success probability increases with the addition of the new substitute model. Consequently, we have:

$$p_{m+1}^{\text{suc}} > p_m^{\text{suc}}.$$

Thus, for any $m > n$, it holds that $p_m^{\text{suc}} > p_n^{\text{suc}}$, meaning that as the number of substitute models increases, the probability of successfully attacking the victim model also increases. This concludes the proof. $\qquad\square$

## F THE PROOF OF THEOREM 2

**Theorem 3.** *Suppose that $p_1, p_2, \ldots, p_w$ be independent and identically distributed (i.i.d.) random variables with cumulative distribution function (CDF) denoted by $F(p)$. Then as $w \rightarrow \infty$, the probability that $\max\{p_1, p_2, \ldots, p_w\} > p_i$ approaches 1 for any $p_i < 1$. Formally,*

$$\lim_{w \to \infty} \Pr\left(\max\{p_1, p_2, \ldots, p_w\} > p_i\right) = 1,$$

$$and \quad \lim_{w \to \infty} \Pr\left(\max\{p_1, p_2, \ldots, p_w\} > \max\{p_1, p_2, \ldots, p_m\}\right) = 1, \quad for \ w > m. \tag{7}$$

*Proof.* We are interested in the probability that the maximum of $w$ i.i.d. random variables exceeds $p_i$. The cumulative distribution function of the maximum of $w$ i.i.d. random variables, $\max\{p_1, p_2, \ldots, p_w\}$, is given by:

$$\Pr\left(\max\{p_1, p_2, \ldots, p_w\} \leq p_i\right) = \Pr\left(p_1 \leq p_i, p_2 \leq p_i, \ldots, p_w \leq p_i\right)$$
$$= [\Pr(p_1 \leq p_i)]^w \tag{8}$$
$$= [F(p_i)]^w.$$

where $F(p_i)$ is the cumulative distribution function evaluated at $p_i$. The probability that $\max\{p_1, p_2, \ldots, p_w\} > p_i$ is the complement of this:

$$\Pr\left(\max\{p_1, p_2, \ldots, p_w\} > p_i\right) = 1 - [F(p_i)]^w. \tag{9}$$

As $w \rightarrow \infty$, the term $[F(p_i)]^w$ tends to 0 for any $p_i < 1$, because $F(p_i) < 1$.

$$\lim_{w \to \infty} [F(p_i)]^w = 0, \ \lim_{w \to \infty} \Pr\left(\max\{p_1, p_2, \ldots, p_w\} > p_i\right) = 1. \tag{10}$$

According to equation equation 10, Thus,

$$\lim_{w \to \infty} \Pr\left(\max\{p_1, p_2, \ldots, p_w\} > \max\{p_1, p_2, \ldots, p_m\}\right.$$
$$= \lim_{w \to \infty} \Pr\left(\max\{\max\{p_1, p_2, \ldots, p_m\}, p_{m+1}, \ldots, p_w\}\right.$$
$$> \max\{p_1, p_2, \ldots, p_m\}$$
$$= 1. \tag{11}$$

Thus, as $w$ increases, the probability that $\max\{p_1, p_2, \ldots, p_w\} > p_i$ approaches 1. Additionlly, when $w > m$, the probability that $\max\{p_1, p_2, \ldots, p_w\} > \max\{p_1, p_2, \ldots, p_m\}$ approaches 1. $\quad\square$

## G FURTHER EXPERIMENTAL RESULTS ON ADDITIONAL DATASETS

Table 7: The attack performance of FLA and other attacks methods on TREC6 and Agnews Datasets. For each metric, the best method is highlighted in **bold** and the runner-up is underlined.

| Method | TREC6 | | | | | | Agnews | | | | | |
| | DistilBERT | | | RoBERTa | | | DistilBERT | | | RoBERTa | | |
| | ASR(%) ↑ | Sim ↑ | Cost ↓ | ASR(%) ↑ | Sim ↑ | Cost ↓ | ASR(%) ↑ | Sim ↑ | Cost ↓ | ASR(%) ↑ | Sim ↑ | Cost ↓ |
|---|---|---|---|---|---|---|---|---|---|---|---|---|
| Bae | 22.40 | 0.761 | 4584 | 21.20 | 0.75 | 4540 | 21.43 | 0.808 | 95288 | 25.64 | 0.776 | 87063 |
| FD | 27.80 | 0.871 | 9686 | 31.40 | 0.873 | 9446 | 38.25 | 0.866 | 228182 | 39.78 | 0.865 | 80864 |
| Hotflip | 38.80 | 0.899 | 3669 | 37.40 | 0.9 | 3605 | 38.25 | 0.842 | 77809 | 37.20 | 0.828 | 61403 |
| PSO | 35.00 | 0.554 | 3233 | 34.20 | 0.927 | 3162 | 34.33 | 0.88 | 63134 | 36.57 | 0.822 | 56569 |
| TextBug | 37.80 | 0.942 | 7771 | 39.60 | 0.957 | 7607 | 43.71 | 0.881 | 142530 | 38.68 | 0.903 | 138905 |
| Leap | 38.66 | 0.886 | 13700 | 39.79 | 0.918 | 13550 | 26.64 | 0.896 | 267596 | 27.39 | 0.92 | 258020 |
| CT-GAT | 10.40 | 0.959 | 5994 | 9.60 | 0.98 | 6009 | 20.36 | 0.918 | 120825 | 34.13 | **0.971** | 111028 |
| HQA | 34.60 | 0.944 | 13297 | 36.00 | 0.946 | 13642 | 30.01 | 0.931 | 220970 | 34.86 | 0.957 | 211105 |
| **FLA** | **42.80** | 0.933 | **0** | **43.60** | 0.935 | **0** | **46.33** | 0.911 | **0** | **41.51** | 0.932 | **0** |

