# OpenReview forum: "No Access, No Safety: Free Lunch Adversarial Attacks on Black-box NLP Models"
_ICLR.cc/2025/Conference — ICLR 2025 Conference Withdrawn Submission_

### Official Review · Reviewer_vxW4 · 2024-10-20

**Soundness:** 2
**Presentation:** 1
**Contribution:** 2
**Rating:** 3
**Confidence:** 4

**Summary:**

This paper discusses a black-box adversarial attack setting for text classification where the attacker does not know the structure nor has access to the victim model. The authors design a new adversarial attack framework called FLA which leverages a set of the victim texts. FLA first does clustering on the victim texts and uses a hierarchical way to train multiple substitution models. For each victim text, it employs multiple adversarial text generation methods and selects the best adversarial example based on the substitution models. If no generated adversarial example satisfies the need, then a new substitution model will be trained. In the paper, the authors also conduct many experiments to show the superiority of their method.

**Strengths:**

1. The paper studies an important topic that is also timely in the era of Large Language Models where the queries to LLMs are restricted.
2. The paper proposes an interesting idea that leverages the unlabeled test set to construct substitution models for generating adversarial texts.

**Weaknesses:**

1. The paper presentation is not good and paper writing is often informal. Specifically, it has the following drawbacks:

a) Some notations are not clearly/correctly defined. For example, Line 184 should be $\hat{y} \in$ instead of $\hat{y} =$; $o_{\hat{y_i}}$ in Eq 6 is not defined; $m$ represents the total number of labels in Line 185 but then is used as the number of substitute models in Theorem 1.

b) Some settings are not clearly stated. For example, Line 406 claims "Based on the analysis in Section 4.1, using two clusters yields the optimal attack performance." However, such analysis is missing in Section 4.1 and it is also not clearly stated that the cluster size is two in Section 4. Besides, the paper claims T5 is the embedding model for FLA but it is unknown which version of T5 is used and how the embedding is calculated. The definition of Cost metric is still confusing in Appendix.

c) Wrong citations. In the paper, Textbugger cites [1] (e.g. Line 102). However, [1] is the PWWS method and [2] should be the right citation for Textbugger.

d) The paper contains confusing sentences and lots of typos. For example, what is "eliciting strong responses" in Line 210? Line 514 is a problematic sentence; Transfer attack (e.g. Line 229) should be **transfer-based attack**; Line 321 should be **Several** ; Line 323 should be **details**; Section 5.4.3 should be "Attacks on LLMs"; Table 4 should be **3** models;

2. The effect of using multiple adversarial text generation methods in FLA is still unclear.

a) First, the paper does not clearly state what attack methods ($M_i$) are used in FLA for Section 5.2.

b) From Table 4, we can find that applying a single attack method in FLA is not always better than applying it directly. For example, using PSO with 7 substitution models in FLA achieves 34.12 ASR on SST5 (DistilBERT),  lower than 45.14 in Table 1. Therefore, it is unclear if the superior performance of FLA in Table 1 is due to the ensemble effect of multiple attack methods.

3. Theorem 1 is not accurate for FLA. According to Lines 790-795, it in fact assumes using the adversarial texts from all substitution models for attacking the victim model and regards the attack succeeds if one of them succeeds. It is different from the workflow of FLA which selects only one for attacking the victim model based on the threshold. As also shown in Table 4, FLA variants (PSO-FLA and TextBug-FLA) using 3 models do not perform better than those using 1 model on the emotional task, which conflicts with Theorem 1.

4. Table 5 lacks adequate baselines. It only uses CT-GAT for comparison. But all previous adversarial attack baselines can be adapted under the transfer-based attack setting. They just need a substitution model with which they can get the necessary information. Then they can directly evaluate the generated adversarial texts from the substitution model on LLMs like GPT4o. So what is the performance of other adversarial attack methods when adapted to the transfer-based attack setting for Table 5?

[1] Ren et al. Generating natural language adversarial examples through probability weighted word saliency

[2] Li et al. TextBugger: Generating Adversarial Text Against Real-world Applications

**Questions:**

1. What is the effect of the embedding model on attack performance of FLA?

2. The authors evaluate the zero-shot data scenario in Section 5.4.2. I assume the public dataset should have ground-truth labels for the texts. I wonder in this case, does FLA method still use clustering to train substitution models? How is it compared with directly training substitution models with ground-truth label?

3. See the weakness 2,3,4.

---

### Official Review · Reviewer_be6P · 2024-10-30

**Soundness:** 2
**Presentation:** 3
**Contribution:** 3
**Rating:** 5
**Confidence:** 4

**Summary:**

This study introduces the Free Lunch Adversarial Attack (FLA), a novel method for executing textual adversarial attacks on NLP models without needing access to the victim model or its training data. By leveraging publicly available pre-trained models to create a shadow dataset, FLA generates substitute models that align with the victim's decision boundaries, significantly enhancing the ASR. Experiments show that FLA achieves up to 45.99% ASR against state-of-the-art LLMs, demonstrating its effectiveness even with limited victim samples. This highlights the serious security risks posed by such black-box attacks on modern NLP systems.

**Strengths:**

- This paper is the first to explore zero-query textual adversarial attacks, achieving an outstanding success rate.
- Unlike existing work that often implements adversarial attacks on BERT series models, the proposed method further demonstrates excellent performance on large language models.

**Weaknesses:**

- This paper makes no contribution to adversarial attack techniques. As stated in Section 4.3, it uses multiple attack methods to generate adversarial samples. Since specific methods are not indicated, I suspect that existing methods are used directly.
- The paper assumes a highly restricted scenario: the attacker cannot access the victim model. While it does not require access to the victim model, it does necessitate the iterative construction of Shadow Datasets and Substitute Models, which is not needed in existing query-based attacks. Therefore, defining the Cost solely as the number of queries to the victim model is unfair, as it overlooks the time spent during the iterative process.
- The evaluation is not comprehensive enough. The paper only evaluates the method on a sentiment analysis task, while the baselines (such as CT-GAT and HQA) often include more NLP tasks. To provide a more thorough evaluation of the proposed method, it should consider incorporating additional tasks.
- In Table 1, $\textit{Sim}$ does not achieve the best performance across all experimental groups. It is well known that $\textit{ASR}$ and Sim should have a trade-off relationship, and the $\textit{ASR}$ of the proposed method may come at the expense of $\textit{Sim}$, which could undermine the contribution of this paper.

**Questions:**

- How does the victim model (including LLMs) perform on the selected dataset in the experiments?
- What specific attack methods are referred to as "Various Attack Methods" in Figure 1? Are they consistent with the baseline?

---

### Official Review · Reviewer_gSnA · 2024-11-04

**Soundness:** 2
**Presentation:** 2
**Contribution:** 2
**Rating:** 3
**Confidence:** 4

**Summary:**

The paper proposes a blackbox adversarial attack method that does not need access to the training data, and knowledge of the victim model. The proposed attacks first builds a pseudo dataset, from a sample of test inputs that the attacker is assumed to have access to. Then it builds a hierarchical set of models to approximate the adversarial decision boundaries. Experimental evaluations show the effectiveness of the proposed attack on 2 datasets SST2 and Emotion.

**Strengths:**

The paper has the following strengths:
* The paper's motivation is clearly described. The use of the test set in the context of adversarial study is interesting.
* The empirical experiments support the effectiveness of the proposed method on 2 datasets.
* The inclusion of evaluation on LLMs is nice and its results are definitely interesting.

**Weaknesses:**

While I generally like the paper's idea, I also have several major concerns:

* The claim "free lunch" is quite overstated. In fact, I would argue that obtaining the test set that's aligned with the distribution of the victim's training set is even more challenging (if not impossible) than either querying the model or obtaining knowledge of the model's architecture.
* Some details are hard to follow. For example, on Line 219: "approximates the distribution of the original training data", I'm not sure how a 2-class dataset follows the distribution of arbitrary original data.
* The clustering step is important, but the intuition and connection to why pseudo-labeling works is not rigorous. Plus, are there any assumptions on the clustering step w.r.t the decision boundaries of the victim's model?
* Evaluation on adversarial defenses is also limited. This state There are various different types of defenses, such as TMD, RanMASK, InfoBert, and AdvFooler, and adversarial training such as FreeLB or ASCC. In fact, I don't agree with this statement "training-free defense methods are more appropriate for FLA".

Nguyen et al. Textual manifold-based defense against natural language adversarial examples. EMNLP'22 \
Hoang et al. Fooling the Textual Fooler via Randomizing Latent Representations. ACL'24 \
Xu et al. 2023. Certified robustness to text adversarial attacks by randomized [MASK].  Computational Linguistics'23. \
Wang et al. InfoBert: Improving robustness of language models from an information theoretic perspective. ICLR'21 \

**Questions:**

Please see weakness section.

---

### Official Review · Reviewer_3qgU · 2024-11-04

**Soundness:** 3
**Presentation:** 2
**Contribution:** 2
**Rating:** 3
**Confidence:** 4

**Summary:**

This paper proposes a novel adversarial attack, named Free Luch Attack (FLA), that can change the decision of the victim model without any access to the model or the dataset. FLA constructs a set of surrogate models by clustering data and iteratively assigning pseudo-labels to the samples. The experiments show that FLA successfully finds adversarial samples with no query to the victim. Furthermore, FLA can also attack LLM, implying its threat in practical applications.

**Strengths:**

- The idea of creating pseudo labels to train surrogate models is novel and interesting.
- FLA can attack the victim model without any access, which is a strong threat model compared to other works
- FLA can also attack LLM, showing that it is threatening to current real-world applications.

**Weaknesses:**

- Other attacks in the experiments have very low success rates and high query costs, which is very strange. In particular, in the HQA paper, they show that HQA decreases the accuracy to a few percent with less than 1000 queries. In [1], TextBugger has more than 70% success rate with avg 300 queries on AGNews, substantially higher than the results in Table 7.
- Since FLA does not know the label of the data, it can not perform targeted adversarial attacks that change the prediction to a target label.
- FLA requires training many surrogate models, which is expensive. Also, it takes FLA similar to universal attacks.


[1] Li, Zongyi, et al. "Searching for an Effective Defender: Benchmarking Defense against Adversarial Word Substitution." Conference on Empirical Methods in Natural Language Processing (EMNLP). 2021.

**Questions:**

- What is the reason for low success rate and high query cost of other attacks?
- What is the training time for FLA?

---

### Note · Authors · 2024-11-15

I have read and agree with the venue's withdrawal policy on behalf of myself and my co-authors.